# Use of the Ball-Cratering Method to Assess the Wear Resistance of a Welded Joint of XAR400 Steel

**DOI:** 10.3390/ma16134523

**Published:** 2023-06-22

**Authors:** Krzysztof Ligier, Mirosław Bramowicz, Sławomir Kulesza, Magdalena Lemecha, Bartosz Pszczółkowski

**Affiliations:** Faculty of Technical Sciences, University of Warmia and Mazury in Olsztyn, M. Oczapowskiego 11, 10-719 Olsztyn, Poland

**Keywords:** low-alloy martensite steels, XAR 400 steel, XRD, GSD function, abrasive wear of welded joints

## Abstract

Wear-resistant steels are designed to allow for operation under extreme loading conditions. They combine large strength with resilience and resistance to abrasive wear. In stock, the steel is subjected to preliminary heat treatment. However, any further processing at temperatures higher than 200 °C results in tempering that influences the mechanical properties of the material. The presented paper aims to study changes in abrasive wear properties across the welded joint made out of this steel, and its prime novelty lies in using the ball-cratering method to test the wear resistance of the joints. To distinguish between different crystalline structures in the weld, metallographic and XRD analyses were performed that resulted in the determination of five primary zones for which wear tests were carried out. Abrasive wear rates, studied across the welded joint, indicate that the material in the HAZ has the lowest resistance to abrasive wear. Similarly, the obtained values of the wear index show decreasing resistance of the material approaching the joint axis.

## 1. Introduction

Low-alloy martensitic steels are the preferred materials for machining parts exposed to abrasive wear [1,2], utilizing their declared workability and weldability together with satisfactory hardness [3,4,5,6]. Supported by Archard’s equation relating wear resistance with hardness, this view has become very popular in the community [7,8,9,10]. However, despite having satisfactory carbon-equivalent values (CEV), such steels undergo unfavorable microstructural changes caused by welding that reduce the durability of machining elements [11,12,13,14,15]. The fact that the welding negatively affects the hardness of high-strength materials is an issue that has been widely investigated in the literature [11,12,13,16]. Other works have demonstrated deterioration in the wear resistance in the weld- and heat-affected zones due to structural changes in these areas [15,17,18]. Welding processes were studied to analyze the details of microstructural modifications within several separate layers of the joints: base material (BM), partially heat-affected zone (PHAZ), heat-affected zone (HAZ), and others [19,20]. Unlike these studies, however, published findings on abrasion resistance of welded joints were of a global character due to the limitations of the research methods. Based on various standards, e.g., ASTM G65, ISO 28080: 2021 (rubber wheel—dry sand), GOST 23.208-79, ASTM G99 and DIN ISO 7148-1 (pin-on-disk) [21,22,23], these measurements were carried out in such a way that they overlapped different zones of the welded joint and hence the determined wear resistance appeared to be somewhat averaged. In order to distinguish between tribological properties of different parts of such a joint, the test area must entirely lie within the zone of interest. Among the methods that meet this requirement is the ball-cratering method.

The ball-cratering method belongs to the microscale abrasive wear tests commonly used to assess the rate of abrasive wear and the wear resistance of the various construction materials: metals [24,25,26], ceramics [27,28], polymers [29,30,31], thin coatings [32,33,34] and others [35]. Its accuracy and effectiveness has resulted in many modifications to investigate, for example, friction and wear processes simultaneously with performed tests [34,36,37,38,39,40]. The ball-cratering method relies on performing small craters within tested materials (even below 0.3 mm in diameter), the shape of which is used to determine several parameters of interest. A small test area makes this method especially useful in studies of materials with a multi-phase microstructure.

The presented work aims to investigate the wear resistance of various zones across a welded joint using the ball-cratering method supported by XRD and surface profile measurements.

## 2. Materials and Methods

### 2.1. Welding Process

The welded joints were made between 10 mm thick sheets of XAR 400 steel (ThyssenKrupp, Duisburg, Germany). The declared chemical composition of this material is shown in Table 1.

Two squares of the steel (150 × 150 mm^2^) were beveled at one side to form a single-V opening for the butt weld (BW 10 V). Conventional MAG 135 multi-run welding (DC+) was carried out in a flat position (PA) using semi-automatic welding machine (KEMPPI KM 400, Lahti, Finland), ESAB OK AristoRod 69 binding agent, and M21 (80% Ar/20% CO_2_) shielding gas. Figure 1 presents a schematic cross section of the obtained multi-run weld. Prior to the welding, the edges were preheated to 50 °C. The weld started with the root stitch (1). Then it was cut from the side of the ridge and the stitch (1a) was laid. In consecutive runs, the two filling stitches (2, 3) and one closing stitch (4) were performed to complete the face of the joint. Details of the welding routine are presented in Table 2.

The performed tensile test of the weld resulted in an *Rm* value equal to 740 MPa. This was well below the yield strength of the base material (1250 MPa). In spite of that, however, the welding appeared to comply with general rules, since values reported elsewhere [41,42] were found to be even lower.

### 2.2. Structural Characterization

Changes in the crystalline structure of the welded XAR 400 steel were investigated via an analysis of XRD spectra using the grain-size distribution (GSD) method. Actually, it is well-known that the Scherrer method is adopted to deal with polydisperse powders in which observed line-broadening occurs due to the finite size of the crystallites, excluding other effects (lattice strain, stacking faults etc.) [42,43,44,45]. To this end, the following definition of the GSD was proposed [42]:(1)$$\mathrm{GSD}\left(x,\rho ,\mu \right)=\frac{x^{\mu }\cdot \exp\left(-x/\rho \right)}{\rho^{\mu +1}\cdot \Gamma \left(\mu +1\right)}$$
where *x* is the grain diameter and *Γ* is the Euler gamma function, while auxiliary variables *ρ* and *µ* are associated with the parameters of interest, i.e., the mean grain diameter *<D>* and size dispersion *σ*, as follows:(2)$$\rho =\frac{\sigma^{2}}{\langle D\rangle }$$
(3)$$\mu =\frac{{\langle D\rangle }^{2}}{\sigma^{2}}-1$$

In his paper [43], Pielaszek demonstrated that the diffraction peak profile for a polydisperse powder can be expressed in terms of given GSD parameters providing that the XRD line is expressed in terms of reciprocal lattice coordinates represented by the scattering vector *q*:(4)$$q=\frac{4\pi }{\lambda }\sin\left(\vartheta \right)$$
where *λ* is the incident beam wavelength and *θ* signifies Bragg’s angle. Example plots of original and converted XRD curves are shown in Figure 2A,B, respectively.

In order to obtain satisfactory accuracy for the GSD method, the line widths must be properly determined. This should happen, for example, with the help of a well-defined profile function fitted to a given XRD peak. To this end, symmetric Pearson VII was usually used because of its flexibility in handling a large variety of tail-shaped curves:(5)$$I\left(q\right)=I_{0}{\left(1+\left(\sqrt[{m}]{2}-1\right){\left(\frac{2\left(q-q_{0}\right)}{w}\right)}^{2}\right)}^{-m}+A\cdot q+B$$
where *q_0_* is the peak centre, *I_0_* is the maximum peak intensity, *w* is peak width, and *m* is shape factor (1—Lorenz, ∞—Gauss). Conversely, *A* and *B* indicate the slope and free term of a linear background. As an example, the least-squares fit of the above function to the (110) XRD peak under investigation is shown in Figure 2B.

In order to solve for two parameters of the GSD distribution from a single XRD line, two independent variables are necessary, e.g., two line widths, measured at various levels of the same XRD peak (possibly near to its top and bottom). To deal with that, various authors proposed to probe the line widths at 20 and 80% of the peak maximum (labelled: *FW_20_* and *FW_80_*, respectively) [42,43,44]. With that information, physically meaningful parameters of the GSD can be finally computed as follows:(6)$$\langle D\rangle =\frac{2\cdot B\cdot C}{FW_{80}}$$
(7)$$\sigma =\frac{2\cdot B\cdot \sqrt{C}}{FW_{80}}$$
where auxiliary variables *A*, *B* and *C* are given as follows:(8)$$A=arcctg\left(\left(2.77069-1.05723\cdot \frac{FW_{20}}{FW_{80}}\right)\cdot 10^{5}\right)$$
(9)$$B=\frac{1.555+8.84\cdot ctg\left(2.237\cdot 10^{-3}-2.101\cdot 10^{3}\cdot A\right)}{10^{3}}$$
(10)$$C=-0.6515-4.63695\cdot 10^{-5}\cdot A$$

For example, a GSD curve obtained following the above procedure is plotted in Figure 2C, where both mean grain size as well as size dispersion are also shown.

In order to verify the results from GSD they can be compared with those obtained using other methods, e.g., volume-weighted average diameter *<D_V_>* defined as follows [44]:(11)$$\langle D_{V}\rangle =\frac{{\langle D\rangle }^{2}+3\sigma^{2}}{\langle D\rangle }$$

Obtained volume-weighted grain diameter is also marked in Figure 2C.

### 2.3. Tribological Tests

Tribological tests were carried out in various zones of the welded joint. To this end, a cross section was prepared and further divided into 3 parts, as shown in Figure 3. The first part covered the weld itself together with the area no further than 23 mm from the axis of the weld, the second one extended to 53 mm from the axis, while the third sample covered the area no further than 82 mm from the axis.

Using preliminary microstructural analysis 5 main zones were distinguished within the cross section of the joint where abrasion resistance tests were performed:Zone 1—base material not affected by the heat (BM)Zone 2—partially heat-affected zone (PHAZ II)Zone 3—partially heat-affected zone (PHAZ I)Zone 4—heat-affected zone (HAZ)Zone 5—root weld zone (WZ I)Zone 6—stitch weld zone (WZ II)

The samples were embedded in acrylic resin and their surface was finished by grinding and polishing. Surface roughness was measured using a Mitutoyo SurfTest profilometer in accordance with the standard ISO 1997. The hardness of the welded joint was determined by the Vickers method in accordance with standard PN-EN ISO 6506 using a Wilson VH1150 hardness tester with a load of 2.94 N, and this method was maintained for 10 s (HV0.3 test scale). Hardness measurements were carried out in 1 mm steps along two lines perpendicular to the weld axis that were 1.7 and 6.4 mm from the bottom of the joined sheet.

### 2.4. Testing Methodology

The study employed the ball-cratering method using a T-20-type tribometer shown in Figure 4. The ball made of 100Cr6 steel and 25.4 mm in diameter (1″) was used as the counter specimen. The hardness of the ball was 58.6 HRC and its surface roughness *R_a_* was 0.177 µm. The test procedures were carried out in accordance with the standard EN-1071-6:2008 with the following parameters:friction assembly load: 0.4 N;counter-sample rotational speed: 150 rpm;sliding distance 179.5 m;test duration: 15 min.

Measurements of abrasive wear were carried out with abrasive slurry made from alundum powder (Al_2_O_3_) with a density of 3.95 g/cm^3^ and sharp-edged grains with a diameter of 3 µm (±1%) (F1200 according to FEPA—Federation of European Producers of Abrasives). The concentration of the slurry was approx. 10% (vol.) and was obtained by mixing 6.4 g Al_2_O_3_ in 100 cm^3^ of distilled water. The slurry was fed onto the friction assembly in an amount of 2 cm^3^/min. The tests were repeated 8 times in each zone with identical test parameters. Before each run, the specimen and the ball were thoroughly cleaned and degreased using ethanol.

The wear rate was assessed based on the average size of the formed craters, which were obtained from the diameters measured along and perpendicular to the rotation axis using a Keyence VHX 7000 digital optical microscope. However, only nearly circular craters were used, for which the two diameters did not differ more than 10% (EN 1071-6:2008). The wear volume was determined using the formula: (12)$$V=\frac{\pi b^{4}}{64\cdot R}$$
where *R* is the radius of the ball and *b* is the average diameter of the crater. In the following, the rate of material wear was determined using the Archard’s equation [8]:(13)$$W_{r}=\frac{V}{S\;N}=\frac{\pi \;b^{4}}{64\;R\;S\;N}$$
where *W_r_* is wear rate, *R* indicates ball radius, *b* is mean crater diameter, *S* is sliding distance, and *N* is normal load.

## 3. Results and Discussion

### 3.1. Roughness Tests Results

Example roughness profile is shown in Figure 5, and corresponding roughness parameters are presented in Table 3.

### 3.2. Hardness Tests Results

The obtained profiles of the hardness values vs. distance from the weld axis shown in Figure 6 reveal significant changes in the mechanical properties of the welded material. Decreased hardness (270 HV0.3) was found right in the weld and in the heat-affected zone compared to the material not subjected to heat. The lowest obtained hardness (164 HV0.3 near the bottom edge of the sample, ca. 5 mm from the axis of the weld) turned out to be significantly lower compared to the base material (420 HV0.3).

Figure 6 shows that the hardness in the central part of the weld was found to be nearly constant at the level of approx. 250HV0.3 regardless of the depth, while the hardness in the heat-affected zone depended on the distance from the edge of the joined material. Close to the root, the hardness in the HAZ decreased and formed a valley centred 5 mm from the axis of the weld due to structural changes within the material [11,19]. A similar effect was not observed in the middle part of the joint, where the hardness only slightly fluctuated over a distance between 10 and 15 mm from the axis and then gradually increased to about 400HV0.3 at a distance 28 mm from the axis. Further, the hardness approached a plateau that varied in the range between 400HV0.3 and 450HV0.3.

### 3.3. Joint Microstructure

Using metallographic analysis (Mi1Fe etchant: 5 mL HNO_3_ + 100 mL C_2_H_5_OH), six specific parts with different microstructural morphology features were distinguished in the investigated area (see Figure 3). Figure 7 shows that cyclic exposure to intense heating caused noticeable changes in the welded joint and the nearby material. Figure 7A exhibits the presence of tempered martensite (TM) with the dispersion precipitates of transition carbides, *ε* and *η*, in the base material. Apart from that, single grains of lower bainite (LB) with plates of supersaturated ferrite can rarely be seen. Elapsed heating is responsible for dissolving transition carbides in the matrix in zone 2 (Figure 7B) and the precipitation of cementite (CC), the coagulation of which is clearly observed in zone 3 (Figure 7C).

Figure 7D reveals that HAZ, marked also as zone 4, contains mainly allotriomorphic ferrite (AF) with finely dispersed pearlite (DP). Figure 7E exhibits structural variety of ferrite inclusions that appear within the joint area, including the root of the filler beads. Figure 7E shows that large amounts of polygonal fine-grained ferrite (FGF) and acicular ferrite (AF), as well as trace amounts of Widmastatten ferrite (WF), can be found in zone 5, which covers the recrystallized weld area. The observed fragmentation of the ferritic structures is caused by repeated recrystallization cycles and rapid heat dissipation. The facing stitch has a morphology typical for welded joints, i.e., anisotropic crystals elongated in the direction of heat dissipation.

### 3.4. Crystalline Structure Analysis

Figure 8 shows a series of XRD spectra with two distinct diffraction peaks, (110) and (200), taken from different parts of the welded joint and nearby material. The diffraction patterns appear similar to those observed in polydisperse solids made from particles of various diameters.

Table 4 lists the GSD parameters together with volume-weighted grain diameters obtained using the procedures presented in the preceding paragraph. It turns out that *<D_V_>* values always exceed those obtained using the GSD method by 20–25%. The observed discrepancies might be due to the spherical, and hence isotropic, morphology of crystallites assumed for the volume-weighted grain size, whereas the strong dependence on the crystallographic orientation actually proves the existence of significant anisotropy in this regard.

Table 4 also lists the results of the shape parameter *m* that defines the specific curvature of the peak profile. Numerical fitting yielded an *m* value in the range from 10 to 14 for the low-angle (110) peak, and in a broader range this extended from 5 to 15 for the high-angle (200) peak. This difference might be due to greater uncertainty in the approximation of the line profile because of the much lower intensity of the (200) peak compared to (110), and thus larger noise amplitude (Figure 9). Apart from that, however, obtained m values much higher than 2 reveal the predominant Gaussian character of the model curve (Pearson VII), which in the following suggests that analyzed XRD peaks are found convoluted multimodal profiles, which agrees well with the polydisperse nature of the crystallites in the material under investigation.

### 3.5. Wear Test Results

Example craters obtained in different parts of the welded joint are shown in Figure 10.

In turn, Figure 11 shows the plot of wear rates determined in different parts of the welded joint. This plot reveals the highest value of the abrasive wear rate in HAZ (zone 4). This gradually decreases with increasing distance from the weld axis, that is, as it moves towards the material least affected by the heat (zones 3, 2 and 1). On the other hand, only small differences in the wear rates between the lower and upper weld areas were found in the joint (zones 5 and 6), about half those observed in zone 4. Such results confirm the previous observations [17,18] that the material in the HAZ is the least resistant to wall wear.

The decreasing abrasion resistance of the welded material is associated with unfavorable microstructural changes and the growth of austenite grains that correspond to the temperature gradients that are responsible for the amount of heat exchanged via thermal conduction and radiative processes. In zones 1 (BM) and 2 (PHAZ II), which are characterized by high hardness, the chasing process, caused by abrasive particles moving over the surface of the sample, dominates (Figure 12A,B). Small dents (microindentation) in the material were also found as a result of the rolling of abrasive particles in the contact area of the sample with the counter-sample. The number of traces of microindentation increases with the decreasing hardness of the material compared to the hardness of the ball [46,47]. In the zones that were most affected by the heat (PHAZ I and HAZ), small amounts of damage were found in the form of tears (Figure 12C,D).

## 4. Conclusions

Obtained results demonstrate that the size of crystallites along [110] and [200] directions increase within recast layer (welded joint), HAZ and PHAZ zones. On the other hand, size dispersion σ is found to be smaller along [200] than [100] axis.Abrasive wear rates in different parts of the welded joint, determined using the ball-cratering method, indicate that the material in the HAZ has the lowest resistance to abrasive wear. The value of wear rate in this area is 2.5 times higher than that in the unwelded material. The obtained results of the wear index indicate a decrease in the resistance of the welded joint material as it approaches the joint axis.The ball-cratering method can be successfully used to evaluate the tribological properties of welded joints in their cross section. Due to small wear trace, it is possible to determine cross-sectional maps of the abrasion resistance of areas of various structural characteristics. Further investigations might be extended into using this method for mapping of the wear rate of the weld joints of other geometries.

## Figures and Tables

**Figure 1 materials-16-04523-f001:**
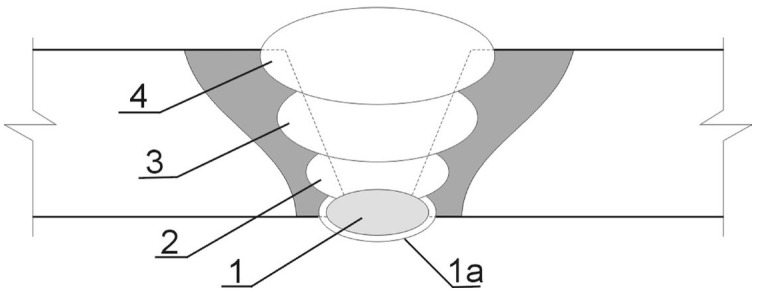
Cross section of a multi-run weld: 1—root stitch, 2,3—filling stitches, 4—closing (facing) stitch.

**Figure 2 materials-16-04523-f002:**
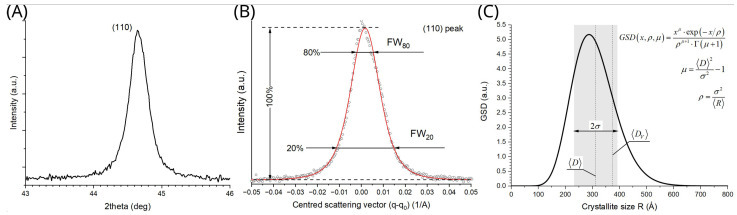
XRD data related to the base material of XAR 400 steel: (**A**) original XRD data—(110) peak in 2theta coordinates, (**B**) least-square fit of Pearson VII function (red line) with linear background (dashed line) to (110) XRD data (open circles), (**C**) plot of resulting GSD function corresponding to the mean grain size *<D>* = 312 nm and size dispersion *σ* = 81 nm (shaded rectangle). For comparison, volume-weighted mean diameter *<D_V_>* = 375 nm is also shown as a vertical dotted line. Note that estimated averages differ from the distribution maximum.

**Figure 3 materials-16-04523-f003:**
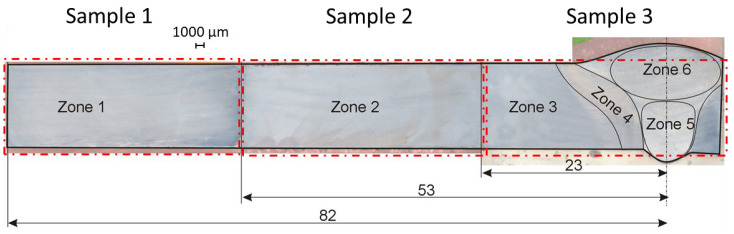
Cross section of the welded joint prepared to tribotests. Red lines mark the boundaries of the zones under investigation.

**Figure 4 materials-16-04523-f004:**
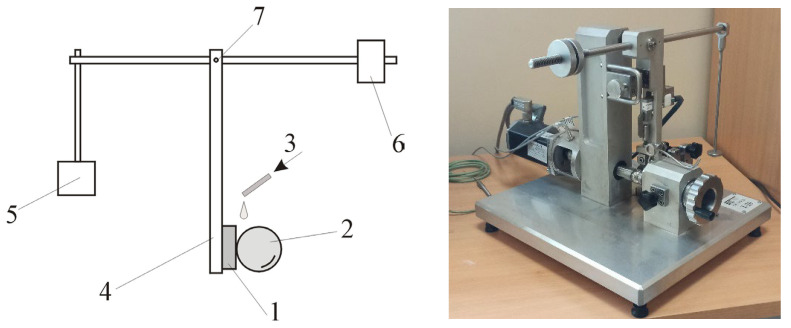
Schematic and general views of the T-20 stand for testing the abrasive wear using the ball-cratering method. 1—sample, 2—ball (counter-sample), 3—abrasive slurry feed, 4—sample holder arm, 5—load, 6—counterweight, 7—pivot.

**Figure 5 materials-16-04523-f005:**
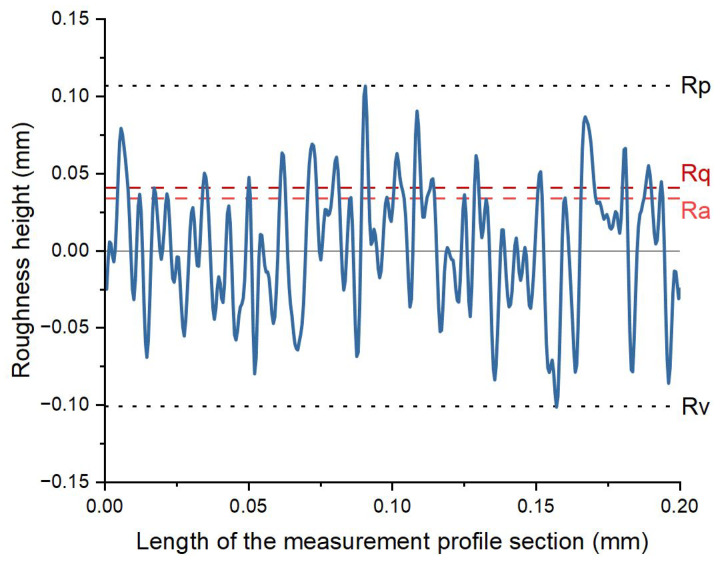
Example roughness profile of sample 1. Dotted lines mark selected roughness measures derived from this profile: Rq (rms profile roughness) and Ra (absolute average profile roughness) as red and orange dashed lines, respectively, and Rp (highest peak) and Rv (deepest valley) as blue dotted lines.

**Figure 6 materials-16-04523-f006:**
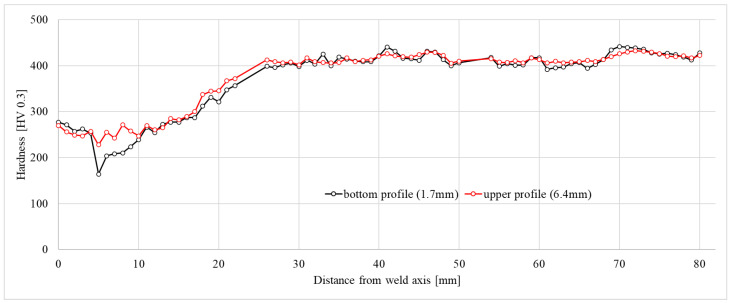
Hardness profile along the welded joint cross section.

**Figure 7 materials-16-04523-f007:**
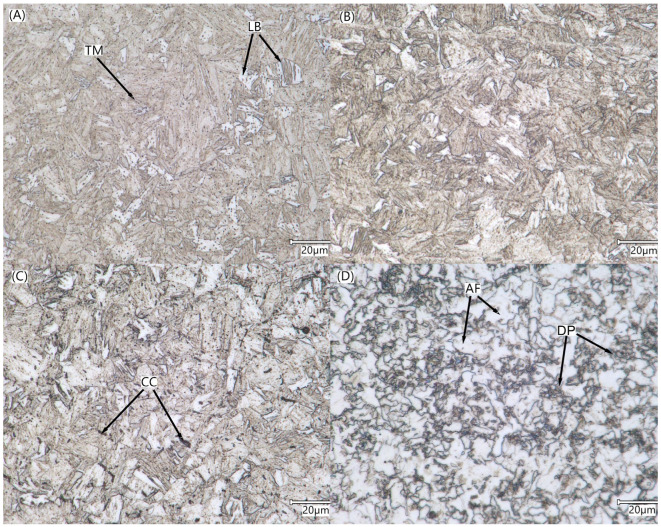
Comparison of microstructural properties observed in different parts of the welded joint: (**A**) BM zone, (**B**) PHAZ-II zone, (**C**) PHAZ-I zone, (**D**) HAZ zone, (**E**) filler stitches (WZ-I), (**F**) facing stitch (WZ-II). Description of the labels: TM—tempered martensite, LB—lower bainite, CC—cementite; AF—allotriomorphic ferrite; FGF—fine-grained ferrite; WF—Widmastatten ferrite.

**Figure 8 materials-16-04523-f008:**
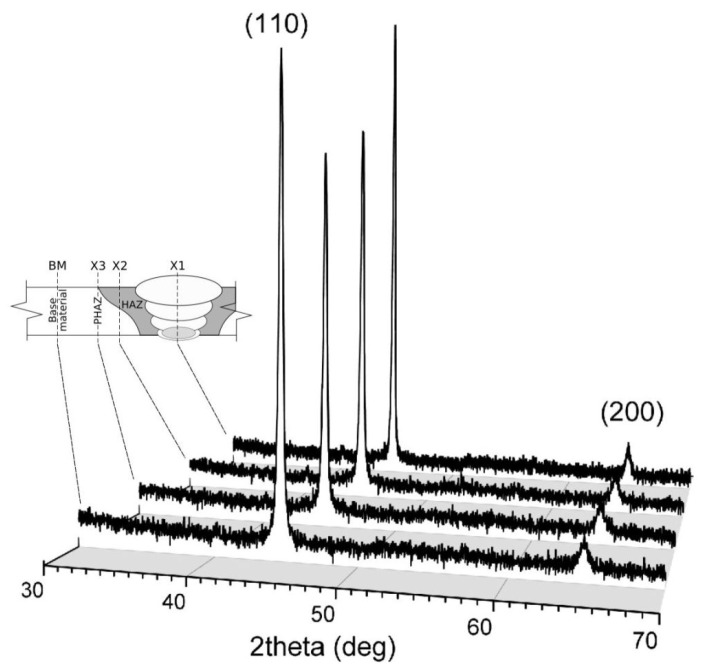
XRD spectra recorded at various parts of the welded joint and nearby material: X1—central axis of the joint, X2—HAZ zone, X3—PHAZ zone, BM—base material.

**Figure 9 materials-16-04523-f009:**
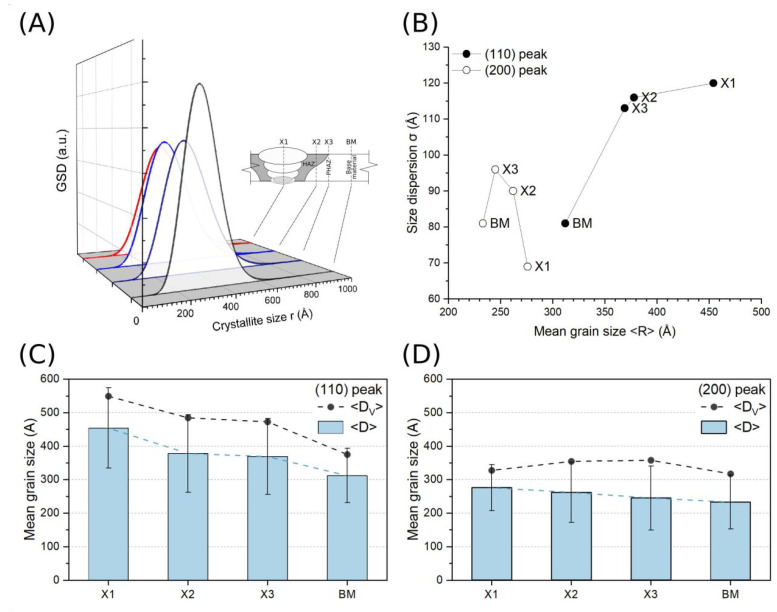
Numerical results of extraction of the mean grain size, dispersion and volume-weighted data using GSD method: (**A**) GSD curves corresponding to different parts of the joint and nearby material, (**B**) plots of the grain size dispersion versus mean grain size determined for two crystallographic directions: [110] (closed circles) and [200] (open circles), (**C**) plots of the mean grain size (bars) and grain size dispersion (error bars) determined from FW20 and FW80 linewidths of (110) peak compared with relevant volume-weighted grain diameters (closed circles), (**D**) plots of the mean grain size (bars) and grain size dispersion (error bars) determined from FW20 and FW80 linewidths of (200) peak compared with relevant volume-weighted grain diameters (closed circles).

**Figure 10 materials-16-04523-f010:**
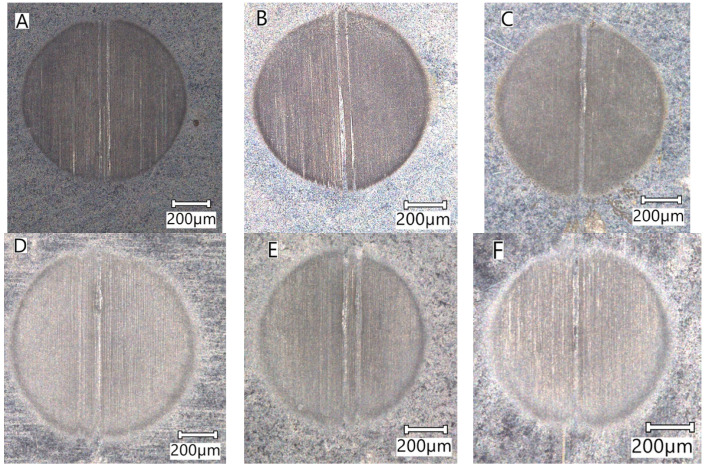
View of sample craters obtained in the tested zones of the welded joint: (**A**) Zone 1 (BM), (**B**) Zone 2 (PHAZ—II), (**C**) Zone 3 (PHAZ—I), (**D**) Zone 4 (HAZ), (**E**) Zone 5 (WZ—I), (**F**) Zone 6 (WZ—II).

**Figure 11 materials-16-04523-f011:**
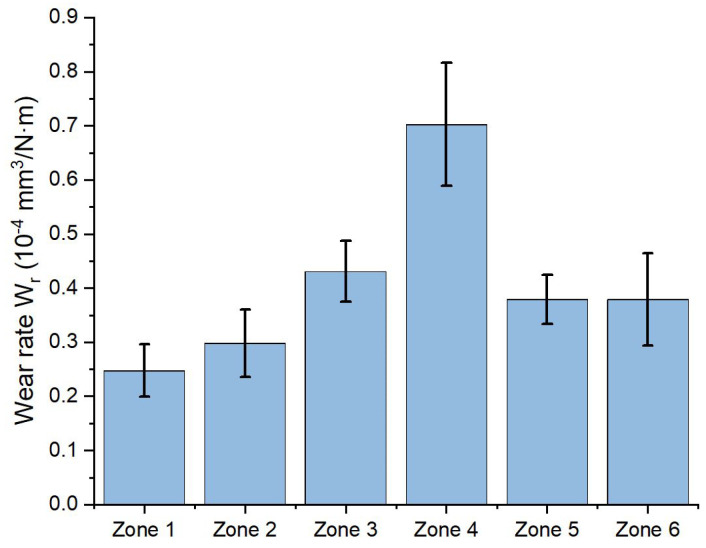
The wear rate values obtained for different zones in the welded joint.

**Figure 12 materials-16-04523-f012:**
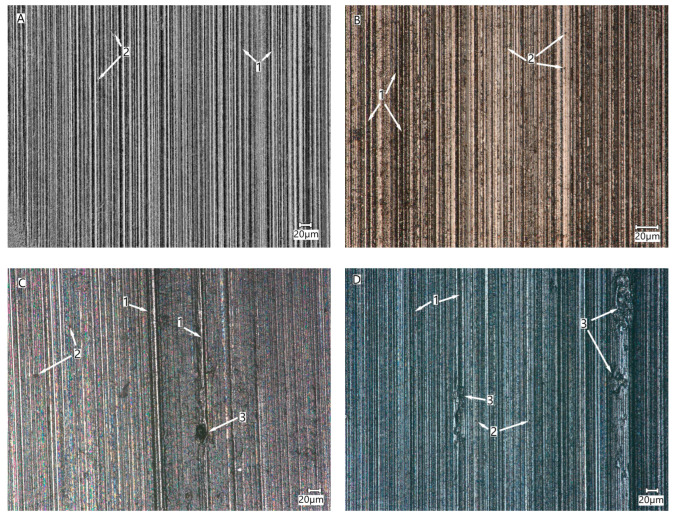
Views of the crater surfaces obtained in different parts of the joint: (**A**) BM, (**B**) PHAZ—II, (**C**) PHAZ—I, (**D**) HAZ. 1—grooves, 2—material dents, 3—rips (tears) of the material.

**Table 1 materials-16-04523-t001:** Declared chemical composition of XAR400 steel.

Declared Chemical Composition [% *w*/*w*]					
C Max.	Si Max.	Mn Max.	Cr Max.	Mo Max.	B Max.
0.2	0.8	1.5	1.0	0.5	0.005

**Table 2 materials-16-04523-t002:** Welding parameters.

Stitch	Current (A)	Voltage (V)	Travel Speed (mm/min)	Gas Flow (l/min)	Heat Input (kJ/mm)
1	105	18.8	96	15	1.2
2	215	28.3	235	15	1.55
3	220	28.7	294	15	1.3
4	215	28.7	265	15	1.4
Shrinkage groove	105	18.8	96	15	1,2

**Table 3 materials-16-04523-t003:** Roughness parameters determined from the roughness profile shown in Figure 5 (to comply with the ISO 4287: 1997 standard).

Profile	R
**λs**	**2.5 µm**
Ra	0.034 µm
Rq	0.041 µm
Rz	0.208 µm

**Table 4 materials-16-04523-t004:** GSD parameters and volume-weighted average sizes of crystallites in the welded joints and nearby material obtained from XRD peaks.

Cross-Sectional Axis	(110)				(200)			
	Mean Grain Diameter *<D>*	Size Dispersion *σ*	Shape Factor *m*	Volume-Weighted Mean Diameter *<D_V_>*	Mean Grain Diameter *<D>*	Size Dispersion *σ*	Shape Factor *m*	Volume-Weighted Mean Diameter *<D_V_>*
	Å	Å	-	Å	Å	Å	-	Å
X1	454	120	13	550	276	69	15	328
X2	378	116	9.8	485	262	90	7.6	355
X3	369	113	9.7	473	245	96	5.5	358
BM	312	81	14	375	233	81	7.4	317

## Data Availability

All information and data are available within the articles.

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
