# Peer review of "Use of the Ball-Cratering Method to Assess the Wear Resistance of a Welded Joint of XAR400 Steel"

_materials, 2023, doi:10.3390/ma16134523_

Round 1

Reviewer 1 Report

The article has a clear and logical structure. Individual chapters have adequate content of text and accompanying images. The authors present things clearly and in an understandable form for the reader. The structure of the article is standard. The authors bring a new perspective to the investigated matter through a qualitative evaluation of the investigated structure using innovative methods. They back up their claims with microstructural analysis, hardness evaluation, and other evaluations. The final evaluations are supported by the achieved results, built on theoretical foundations. After studying the article, I have several formal comments that I would like to incorporate. Overall, I recommend the article for publication, after incorporating the comments. Comments on the article: - in equations, it is desirable to use the exponential form of numbers, e.g. equation 9: 0.002237 and so on - in the article, please unify the labeling of quantities in the same style: e.g. description of figure 2: correct use of italic font for marking constants, etc. - figure 5: the indication of the displayed quantity is missing on the individual axes. Only her units are listed here, and even these are poorly visible. at the same time, it is advisable to add either a trend line or a calculated value /median, average, etc./ to this graph. - picture 6: please add to the graph the points /functional values/ from which the graph was constructed. in the graph description there is a typo /uper profile/ -figure 11: please remove the exponential shape of the numbers from the graph, it is distracting. I recommend using the following description on the vertical axis: wr [umm3/N.m], then on the vertical axis the values will be 0.1 ....0.9

In the presented article, the authors focus on the study of use of the ball-cratering method to assess the wear resistance of a welded joint of XAR400 steel. Low-alloy martensitic steels are mainly of interest. The authors are primarily devoted to studying changes in abrasive wear properties across the welded joint made in this steel. They carried out analyses that resulted in the determination of five primary zones for which wear tests. In the article, from the point of view of the theoretical background, they describe welding process, structural characterization, tribological tests and testing methodology. The mentioned methods and procedures are supplemented with mathematical formulas, with relevant references to the used literature. The authors start from known relationships, while directly using these for the evaluation of the investigated structure, which is XAR 400 steel. As part of tribological tests, the weld in question is divided into several zones, which are subsequently examined. From the point of view of methodology and structure, this part of the contribution is clear and understandable, with adequate informational value for the reader. Furthermore, this knowledge is supplemented by the achieved results, where the authors evaluate: roughness, hardness, joint microstructure, and crystalline structure. The article also contains an assessment from the point of view of wear analysis. The original results of the authors are documented for each research method used. It is mainly about graphic processing, in the sense of evaluation of hardness according to Vickers, example of roughness, together with the given parameters. Images from microstructural analysis are shown to show microscopic views of the individual investigated weld zones. Here, the authors point out individual salient features by marking them directly in the given images. Table 3 further summarizes selected GSD parameters and other quantities obtained from XRD peaks. In the end, samples of craters obtained in the tested zones of the welded joint are presented. In the end, the authors briefly evaluate the achieved results. The structure of the article is logical and balanced. From the point of view of innovativeness, the focus of the chapter is on the achieved results and the discussions related to them. The overall evaluation in chapter 4 is clear, but it can be supplemented with expanding conclusions, or about the vision of the authors in scientific work in the near future. Based on the study of the article /and together with the comments/, I recommend the article for acceptance, after incorporating the comments.

Reviewer 2 Report

In this manuscript, the authors used ball-cratering method to assess the were resistance of localized zones of the welded joint of XAR400 steel. Some interesting results were presented. However, some important information for the future readers is missing. And the following issues must be addressed before further proceeding:

1.      The English writing need to be improved. The Abstract section is relatively concise. The most significant results and conclusion should be presented in the Abstract section.

2.      Apart from the mentioned ball-cratering method, it is suggested to list several methods that are commonly used to assess tribological properties for weld joints and describe their drawbacks.

3.      In the Abstract and Introduction section, the authors should highlight the novelty of this study.

4.      In the Introduction section the authors stated ‘That the welding negatively affects the hardness of high strength materials is an issue widely investigated in the literature’. To better support this claim, the authors can refer to some latest papers: doi.org/10.1007/s00170-022-09793-x; doi.org/10.1016/j.matdes.2022.111492

5.      The detailed welding parameters can be presented, and the filling materials composition should also be given.

6.      For Figure 5, what dose the X axis represent? Please indicate in the figure.

7.      The information of the etchant to prepare the samples for metallographic analysis is missing.

8.      Scale bar is missing in Figure 3.

9.      For the ball-cratering test, what is the size of the tested sample? Different parts of the joints (ie. BM zone, PHAZ-II zone, PHAZ-I zone, HAZ zone, …….) were separately sectioned from the whole weld joint?

Need to be improved.

Reviewer 3 Report

I find the manuscript entitled "Use of the ball-cratering method to assess the wear resistance of a welded joint of XAR400 steel " well prepared, efficiently organized,  and presenting an in-depth analysis of the samples behavior during wear tests conducted through ball-cratering method.

My congratulations to the authors come with only some small suggestions: since the abstract should be a self-standing text, usually a small hint/part of the conclusion/results obtained should be provided in the abstract.     I noticed some ASTM standards are mentioned. And afterwards some ISO standards were used. Since ASTM standards are commonly used in the US, the authors have knowledge of  similar/corresponding ISO standards? It's not necessarily wrong, but if we have ISO standards we usually tend to use those.

Some adjustments still needs to be done in the references. I'm sure it will be taken care during the final check of the manuscript.

I find the manuscript entitled "Use of the ball-cratering method to assess the wear resistance of a welded joint of XAR400 steel " well prepared, efficiently organized,  and presenting an in-depth analysis of the samples behavior during wear tests conducted through ball-cratering method.

My congratulations to the authors come with only some small suggestions: since the abstract should be a self-standing text, usually a small hint/part of the conclusion/results obtained should be provided in the abstract.     I noticed some ASTM standards are mentioned. And afterwards some ISO standards were used. Since ASTM standards are commonly used in the US, the authors have knowledge of  similar/corresponding ISO standards? It's not necessarily wrong, but if we have ISO standards we usually tend to use those.

Some adjustments still needs to be done in the references. I'm sure it will be taken care during the final check of the manuscript.

Round 2

Reviewer 2 Report

It can be accepted now.